# Improved Multimodal Deep Learning with Variation of Information

**Kihyuk Sohn, Wenling Shang and Honglak Lee**
University of Michigan Ann Arbor, MI, USA
{kihyuks,shangw,honglak}@umich.edu

## Abstract

Deep learning has been successfully applied to multimodal representation learning problems, with a common strategy to learning joint representations that are shared across multiple modalities on top of layers of modality-specific networks. Nonetheless, there still remains a question how to learn a good association between data modalities; in particular, a good generative model of multimodal data should be able to reason about missing data modality given the rest of data modalities. In this paper, we propose a novel multimodal representation learning framework that explicitly aims this goal. Rather than learning with maximum likelihood, we train the model to minimize the *variation of information*. We provide a theoretical insight why the proposed learning objective is sufficient to estimate the data-generating joint distribution of multimodal data. We apply our method to restricted Boltzmann machines and introduce learning methods based on contrastive divergence and multi-prediction training. In addition, we extend to deep networks with recurrent encoding structure to finetune the whole network. In experiments, we demonstrate the state-of-the-art visual recognition performance on MIR-Flickr database and PASCAL VOC 2007 database with and without text features.

## 1 Introduction

Different types of multiple data modalities can be used to describe the same event. For example, images, which are often represented with pixels or image descriptors, can also be described with accompanying text (e.g., user tags or subtitles) or audio data (e.g., human voice or natural sound). There have been several applications of multimodal learning from multiple domains such as emotion [13] and speech [10] recognition with audio-visual data, robotics applications with visual and depth data [15, 17, 32, 23], or medical applications with visual and temporal data [26]. These data from multiple sources are *semantically* correlated, and sometimes provide complementary information to each other. In order to exchange such information, it is important to capture a high-level association between data modalities with a compact set of latent variables. However, learning associations between multiple *heterogeneous* data distributions is a challenging problem.

A naive approach is to concatenate the data descriptors from different sources of input to construct a single high-dimensional feature vector and use it to solve a unimodal representation learning problem. Unfortunately, this approach has been unsuccessful since the correlation between features in each data modality is much stronger than that between data modalities [21]. As a result, the learning algorithms are easily tempted to learn dominant patterns in each data modality separately while giving up learning patterns that occur simultaneously in multiple data modalities. To resolve this issue, deep learning methods, such as deep autoencoders [9] or deep Boltzmann machines (DBM) [24], have been used to this problem [21, 27], with a common strategy to learning joint representations that are shared across multiple modalities at the higher layer of the deep network after learning layers of modality-specific networks. The rationale is that the learned features may have less within-modality correlation than raw features, and this makes it easier to capture patterns across data modalities. Despite the promise, there still remains a challenging question how to learn a good association between multiple data modalities that can effectively deal with missing data modalities in the testing time.

One necessary condition of being a good generative model of multimodal data is to have an ability to predict or reason about missing data modalities given partial observation. To this end, we propose

a novel multimodal representation learning framework that explicitly aims this goal. The key idea is to minimize the information distance between data modalities through the shared latent representations. More concretely, we train the model to minimize the *variation of information* (VI), an information theoretic measure that computes the distance between random variables, i.e., multiple data modalities. Note that this is in contrast to the previous approaches on multimodal deep learning, which are based on maximum (joint) likelihood (ML) learning [21, 27]. We provide an intuition how our method could be more effective in learning the joint representation of multimodal data than ML learning, and show theoretical insights why the proposed learning objective is sufficient to estimate the data-generating joint distribution of multimodal data. We apply the proposed framework to multimodal restricted Boltzmann machine (MRBM). We introduce two learning algorithms, based on contrastive divergence [19] and multi-prediction training [6]. Finally, we extend to multimodal deep recurrent neural network (MDRNN) for unsupervised finetuning of whole network. In experiments, we demonstrate the state-of-the-art visual recognition performance on MIR-Flickr database and PASCAL VOC 2007 database with and without text features.

## 2 Multimodal Learning with Variation of Information

In this section, we propose a novel training objective based on the VI. We make a comparison to the ML objective, a typical learning objective for training models of multimodal data, to give an insight how our proposal outperforms the baseline. Finally, we establish a theorem showing that the proposed learning objective is sufficient to obtain a good generative model that fully recovers the joint data-generating distribution of multimodal data.

**Notation.** We use uppercase letters $X, Y$ to denote random variables, lowercase letters $x, y$ for realizations. Let $P_{\mathcal{D}}$ be the data-generating distribution and $P_\theta$ the model distribution parameterized by $\theta$. For presentation clarity, we slightly abuse the notation for $Q$ to denote conditional $(Q(x|y), Q(y|x))$, marginal $(Q(x), Q(y))$, as well as joint distributions $(Q(x, y))$ that are derived from the joint distribution $Q(x, y)$. The type of distribution for $Q$ should be clear from the context.

### 2.1 Minimum Variation of Information Learning

Motivated from the necessary condition of good generative models to reason about the missing data modality, it seems natural to learn to maximize the amount of information that one data modality has about the others. We quantify such an amount of information between data modalities using variation of information (VI). The VI is an information theoretic measure that computes the information distance between two random variables (e.g., data modalities), and is written as follows:[1]

$$\mathrm{VI}_Q(X, Y) = -\mathbb{E}_{Q(X,Y)}\big[\log Q(X|Y) + \log Q(Y|X)\big] \tag{1}$$

where $Q(X, Y) = P_\theta(X, Y)$ is any joint distribution on random variables $(X, Y)$ parametrized by $\theta$. Informally, VI is small when the conditional LLs $Q(X|Y)$ and $Q(Y|X)$ are "peaked", meaning that $X$ has low entropy conditioned on $Y$ and vice versa. Following the intuition, we define new multimodal learning criteria, a *minimum variation of information* (MinVI) learning, as follows:

$$\textbf{\textit{MinVI:}} \quad \min_\theta \mathcal{L}^{\mathrm{VI}}(\theta), \quad \mathcal{L}^{\mathrm{VI}}(\theta) = -\mathbb{E}_{P_{\mathcal{D}}(X,Y)}\big[\log P_\theta(X|Y) + \log P_\theta(Y|X)\big] \tag{2}$$

Note the difference in $\mathcal{L}^{\mathrm{VI}}(\theta)$ that we take the expectation over $P_{\mathcal{D}}$ in $\mathcal{L}^{\mathrm{VI}}(\theta)$. Furthermore, we observe that the MinVI objective can be decomposed into a sum of two negative conditional LLs. This indeed well aligns with our initial motivation about reasoning missing data modality. In the following, we provide a more insight of our MinVI objective in relation to the ML objective, which is a standard learning objective in generative models.

### 2.2 Relation to Maximum Likelihood Learning

The ML objective function can be written as a minimization of the negative LL (NLL) as follows:

$$\textbf{\textit{ML:}} \quad \min_\theta \mathcal{L}^{\mathrm{NLL}}(\theta), \quad \mathcal{L}^{\mathrm{NLL}}(\theta) = -\mathbb{E}_{P_{\mathcal{D}}(X,Y)}\big[\log P_\theta(X, Y)\big], \tag{3}$$

and we can show that the NLL objective function is reformulated as follows:

$$2\mathcal{L}^{\mathrm{NLL}}(\theta) = \underbrace{KL\left(P_{\mathcal{D}}(X)\|P_\theta(X)\right) + KL\left(P_{\mathcal{D}}(Y)\|P_\theta(Y)\right)}_{(a)} +$$

$$\underbrace{\mathbb{E}_{P_{\mathcal{D}}(X)}\big[KL\left(P_{\mathcal{D}}(Y|X)\|P_\theta(Y|X)\right)\big] + \mathbb{E}_{P_{\mathcal{D}}(Y)}\big[KL\left(P_{\mathcal{D}}(X|Y)\|P_\theta(X|Y)\right)\big]}_{(b)} + C, \tag{4}$$

where $C$ is a constant which is irrelevant to $\theta$. Note that (b) is equivalent to $\mathcal{L}^{\mathrm{VI}}(\theta)$ in Equation (2) up to a constant. We provide a full derivation of Equation (4) in supplementary material.

Ignoring the constant, the NLL objective is composed of four terms of KL divergence. Since KL divergence is non-negative and is $0$ only when two distributions match, the ML learning in Equation (3) can be viewed as a distribution matching problem involving (a) marginal likelihoods and (b) conditional likelihoods. Here, we argue that (a) is more difficult to optimize than (b) because there are often too many modes in the marginal distribution. Compared to that, the number of modes can be dramatically reduced in the conditional distribution since the conditioning variables may restrict the support of random variable effectively. Therefore, (a) may become a dominant factor to be minimized during the optimization process and as a trade-off, (b) will be easily compromised, which makes it difficult to learn a good association between data modalities. On the other hand, the MinVI objective focuses on modelling the conditional distributions (Equation (4)), which is arguably easier to optimize. Indeed, similar argument has been made for generalized denoising autoencoders (DAEs) [1] and generative stochastic networks (GSNs) [2], which focus on learning the transition operators (e.g., $P_\theta(X|\tilde{X})$, where $\tilde{X}$ is a corrupted version of data $X$, or $P_\theta(X|H)$, where $H$ can be arbitrary latent variables) to bypass an intractable problem of learning density model $P_\theta(X)$.

## 2.3 Theoretical Results

Bengio et al. [1, 2] proved that learning transition operators of DAEs or GSNs is sufficient to learn a good generative model that estimates a data-generating distribution. Under similar assumptions, we establish a theoretical result that we can obtain a good density estimator for joint distribution of multimodal data by learning the transition operators derived from the conditional distributions of one data modality given the other. In multimodal learning framework, the transition operators $T_n^{\mathcal{X}}$ and $T_n^{\mathcal{Y}}$ with model distribution $P_{\theta_n}(X,Y)$ are defined for Markov chains of data modalities $X$ and $Y$, respectively. Specifically, $T_n^{\mathcal{X}}(x[t]|x[t-1]) = \sum_{y\in\mathcal{Y}} P_{\theta_n}(x[t]|y) P_{\theta_n}(y|x[t-1])$ and $T_n^{\mathcal{Y}}$ is defined in a similar way. Now, we formalize the theorem as follows:

**Theorem 2.1** *For finite state space $\mathcal{X},\mathcal{Y}$, if, $\forall x \in \mathcal{X}, \forall y \in \mathcal{Y}$, $P_{\theta_n}(\cdot|y)$ and $P_{\theta_n}(\cdot|x)$ converges in probability to $P_{\mathcal{D}}(\cdot|y)$ and $P_{\mathcal{D}}(\cdot|x)$, respectively, and $T_n^{\mathcal{X}}$ and $T_n^{\mathcal{Y}}$ are ergodic Markov chains, then, as the number of examples $n \to \infty$, the asymptotic distribution $\pi_n(X)$ and $\pi_n(Y)$ converge to data-generating marginal distributions $P_{\mathcal{D}}(X)$ and $P_{\mathcal{D}}(Y)$, respectively. Moreover, the joint probability distribution $P_{\theta_n}(x,y)$ converges to $P_{\mathcal{D}}(x,y)$ in probability.*

The proof is provided in supplementary material. The theorem ensures that the MinVI objective can lead to a good generative model estimating the joint data-generating distribution of multimodal data. The theorem holds under two assumptions, consistency of density estimators and ergodicity of transition operators. The ergodicity of transition operators are satisfied for wide variety of neural networks, such as an RBM or DBM. [2] The consistency assumption is more difficult to satisfy and the aforementioned deep energy-based models nor RNN may not satisfy the condition due to the approximated posteriors using factorized distribution. Probably, deep networks that allow exact posterior inference, such as stochastic feedforward neural networks [20, 29], could be a better model in our multimodal learning framework, but we leave this as a future work.

# 3 Application to Multimodal Deep Learning

In this section, we describe the MinVI learning in multimodal deep learning framework. To overview our pipeline, we use the commonly used network architecture that consists of layers of modality-specific deep networks followed by a layer of neural network that jointly models the multiple modalities [21, 27]. The network is trained in two steps: In layer-wise pretraining, each layer of modality-specific deep network is trained using restricted Boltzmann machines (RBMs). For the top-layer shared network, we train MRBM with MinVI objective (Section 3.2). Then, we finetune the whole deep network by constructing multimodal deep recurrent neural network (MDRNN) (Section 3.3).

## 3.1 Restricted Boltzmann Machines for Multimodal Learning

The restricted Boltzmann machine (RBM) is an undirected graphical model that defines the distribution of visible units using hidden units. For multimodal input, we define the joint distribution of

multimodal RBM (MRBM) [21, 27] as $P(x, y, h) = \frac{1}{Z} \exp\left(-E(x, y, h)\right)$ with the energy function:

$$E(x, y, h) = -\sum_{i=1}^{N_x} \sum_{k=1}^{K} x_i W_{ik}^x h_k - \sum_{j=1}^{N_y} \sum_{k=1}^{K} y_j W_{jk}^y h_k - \sum_{k=1}^{K} b_k h_k - \sum_{i=1}^{N_x} c_i^x x_i - \sum_{j=1}^{N_y} c_j^y y_j, \quad (5)$$

where $Z$ is the normalizing constant, $x \in \{0,1\}^{N_x}$, $y \in \{0,1\}^{N_y}$ are the binary visible (i.e., observation) variables of multimodal input, and $h \in \{0,1\}^K$ are the binary hidden (i.e., latent) variables. $W^x \in \mathbb{R}^{N_x \times K}$ defines the weights between $x$ and $h$, and $W^y \in \mathbb{R}^{N_y \times K}$ defines the weights between $y$ and $h$. $c^x \in \mathbb{R}^{N_x}$, $c^y \in \mathbb{R}^{N_y}$, and $b \in \mathbb{R}^K$ are bias vectors corresponding to $x$, $y$, and $h$, respectively. Note that the MRBM is equivalent to an RBM whose visible variables are constructed by concatenating the visible variables of multiple input modalities, i.e., $v = [x\,;y]$.

Due to bipartite structure, variables in the same layer are conditionally independent given the variables of the other layer, and the conditional probabilities are written as follows:

$$P(h_k = 1 \mid x, y) = \sigma\left(\sum_i W_{ik}^x x_i + \sum_j W_{jk}^y y_j + b_k\right), \quad (6)$$

$$P(x_i = 1 \mid h) = \sigma\left(\sum_k W_{ik}^x h_k + c_i^x\right), \quad P(y_j = 1 \mid h) = \sigma\left(\sum_k W_{jk}^y h_k + c_j^y\right), \quad (7)$$

where $\sigma(x) = \frac{1}{1+\exp(-x)}$. Similarly to the standard RBM, the MRBM can be trained to maximize the joint LL ($\log P(x, y)$) using stochastic gradient descent (SGD) while approximating the gradient with contrastive divergence (CD) [8] or persistent CD (PCD) [30]. In our case, however, we train the MRBM in MinVI criteria. We will discuss the inference and training algorithms in Section 3.2.

When we have access to all data modalities, we can use Equation (6) for exact posterior inference. On the other hand, when some of the input modalities are missing, the inference is intractable, and we resort to the variational method. For example, when we are given $x$ but no $y$, the true posterior can be approximated with a fully factorized distribution $Q(y, h) = \prod_j \prod_k Q(y_j)Q(h_k)$ by minimizing the $KL\left(Q(y, h)\|P_\theta(y, h|x)\right)$. This leads to the following fixed-point equations:

$$\hat{h}_k = \sigma\left(\sum_i W_{ik}^x x_i + \sum_j W_{jk}^y \hat{y}_j + b_k\right), \quad \hat{y}_j = \sigma\left(\sum_k W_{jk}^y \hat{h}_k + c_j^y\right), \quad (8)$$

where $\hat{h}_k = Q(h_k)$ and $\hat{y}_j = Q(y_j)$. The variational inference proceeds by alternately updating the mean-field parameters $\hat{h}$ and $\hat{y}$ that are initialized with all 0's.

### 3.2 Training Algorithms

**CD-PercLoss.** As in Equation (2), the objective function can be decomposed into two conditional LLs, and the MRBM with MinVI objective can be trained equivalently by training the two conditional RBMs (CRBMs) while sharing the weights. Since the objective functions are the sum of two conditional LLs, we compute the (approximate) gradient of each CRBM separately using CD-PercLoss [19] and accumulate them to update parameters.[3]

**Multi-Prediction.** We found a few practical issues of CD-PercLoss training: First, the gradient estimates are inaccurate. Second, there exists a difference between encoding process of training and testing, especially when the unimodal query (e.g., one of the data modality is missing) is considered for testing. As an alternative objective, we propose multi-prediction (MP) training of MRBM in MinVI criteria. The MP training was originally proposed to train deep Boltzmann machines (DBMs) [6] as an alternative to the stochastic approximation procedure learning [24]. The idea is to train the model good at predicting any subset of input variables given the rest of them by constructing the recurrent network with encoding function derived from the variational inference problem.

The MP training can be adapted to train MRBM with MinVI objective with some modifications. For example, the CRBM with an objective $\log P(y|x)$ can be trained by randomly selecting the subset of variables to be predicted only from the target modality $y$, but the conditioning modality $x$

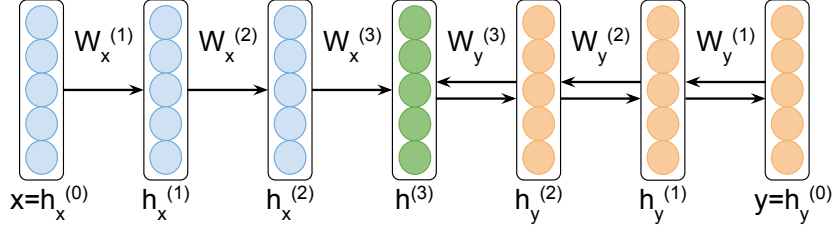

Figure 1: An instance of MDRNN with target $y$ given $x$. Multiple iterations of bottom-up updates ($y \to h^{(3)}$; Equation (11)) and top-down updates ($h^{(3)} \to y$; Equation (13)) are performed. The arrow indicates encoding direction.

is assumed to be given in all cases. Specifically, given an arbitrary subset $s \subset \{1, \cdots, N_y\}$ drawn from the independent Bernoulli distribution $P_S$, the MP algorithm predicts $y_s = \{y_j : j \in s\}$ given $x$ and $y_{\backslash s} = \{y_j : j \notin s\}$ through the iterative encoding function derived from fixed-point equations

$$\hat{h}_k = \sigma\Big(\sum_i W_{ik}^x x_i + \sum_{j \in s} W_{jk}^y \hat{y}_j + \sum_{j \notin s} W_{jk}^y y_j + b_k\Big), \ \hat{y}_j = \sigma\Big(\sum_k W_{jk}^y \hat{h}_k + c_j^y\Big), j \in s, \quad (9)$$

which is a solution to the variational inference problem $\min_Q KL\left(Q(y_s, h) \| P_\theta(y_s, h | x, y_{\backslash s})\right)$ with factorized distribution $Q(y_s, h) = \prod_{j \in s} \prod_k Q(y_j) Q(h_k)$. Note that Equation (9) is similar to the Equation (8) except that only $y_j, j \in s$ are updated. Using an iterative encoding function, the network parameters are trained using SGD while computing the gradient by backpropagating the error between the prediction and the ground truth of $y_s$ through the derived recurrent network. The MP formulation (e.g., encoding function) of the CRBM with $\log P(x|y)$ can be derived similarly, and the gradients are simply the addition of two gradients that are computed individually.

We have two additional hyper parameters, the number of mean-field updates and the sampling ratio of a subset $s$ to be predicted from the target data modality. In our experiments, it was sufficient to use $10 \sim 20$ iterations until convergence. We used the sampling ratio of 1 (i.e., all the variables in the target data modality are to be predicted) since we are already conditioned on one data modality, which is sufficient to make a good prediction of variables in the target data modality.

### 3.3 Finetuning Multimodal Deep Network with Recurrent Neural Network

Motivated from the MP training of MRBM, we propose multimodal deep recurrent neural network (MDRNN) that tries to predict the target modality given the input modality through the recurrent encoding function, which iteratively performs a full pass of bottom-up and top-down encoding from bottom-layer visible variables to top-layer joint representation back to bottom-layer through the modality-specific deep networks. We show an instance of $L = 3$ layer MDRNN in Figure 1, and the encoding functions are written as follows:[4]

$$x \to h_x^{(L-1)}: \quad h_x^{(l)} = \sigma\left(W^{x,(l)\top} h_x^{(l-1)} + b^{x,(l)}\right), l = 1, \cdots, L-1 \quad (10)$$

$$y \to h_y^{(L-1)}: \quad h_y^{(l)} = \sigma\left(W^{y,(l)\top} h_y^{(l-1)} + b^{y,(l)}\right), l = 1, \cdots, L-1 \quad (11)$$

$$h_x^{(L-1)}, h_y^{(L-1)} \to h^{(L)}: \quad h^{(L)} = \sigma\left(W^{x,(L)\top} h_x^{(L-1)} + W^{y,(L)\top} h_y^{(L-1)} + b^{(L)}\right) \quad (12)$$

$$h^{(L)} \to y: \quad h_y^{(l-1)} = \sigma\left(W^{y,(l)} h_y^{(l)} + b^{y,(l-1)}\right), l = L, \cdots, 1 \quad (13)$$

where $h_x^{(0)} = x$ and $h_y^{(0)} = y$. The visible variables of the target modality are initialized with 0's. In other words, in the initial bottom-up update, we compute $h^{(L)}$ only from $x$ while setting $y = 0$ using Equation (10),(11),(12). Then, we run multiple iterations of top-down (Equation (13)) and bottom-up updates (Equation (11), (12)). Finally, we compute the gradient by backpropagating the reconstruction error of target modality through the network.

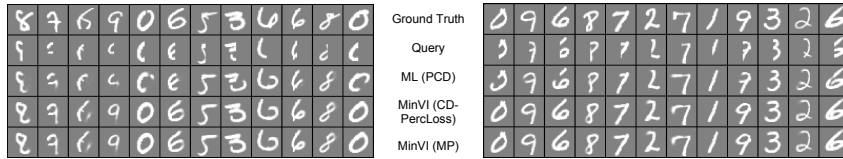

Figure 2: Visualization of samples with inferred missing modality. From top to bottom, we visualize ground truth, left or right halves of digits, generated samples with inferred missing modality using MRBM with ML objective, MinVI objective using CD-PercLoss and MP training methods.

| Input modalities at test time | Left+Right | Left | Right |
|---|---|---|---|
| ML (PCD) | **1.57**% | 14.98% | 18.88% |
| MinVI (CD-PercLoss) | 1.71% | 9.42% | 11.02% |
| MinVI (MP) | 1.73% | **6.58**% | **7.27**% |

Table 1: Test set handwritten digit recognition errors of MRBMs trained with different objectives and learning algorithms. Linear SVM was used for classification with joint feature representations.

## 4 Experiments

### 4.1 Toy Example on MNIST

In our first experiment, we evaluate the proposed learning algorithm on the MNIST handwritten digit recognition dataset [16]. We consider left and right halves of the digit images as two input modalities and report the recognition performance with different combinations of input modalities at the test time, such as full (left + right) or missing (left or right) data modalities. We compare the performance of the MRBM trained with 1) ML objective using PCD [30], or MinVI objectives with 2) CD-PercLoss or 3) MP training. The recognition errors are provided in Table 1. Compared to ML training, the recognition errors for unimodal queries are reduced by more than a half with MP training of MinVI objective. For multimodal queries, the model trained with ML objective performed the best, although the performance gain was incremental. CD-PercLoss training of MinVI objective also showed significant improvement over ML training, but the errors were not as low as those obtained with MP training. We believe that, although it is an approximation of MinVI objective, the exact gradient for MP algorithm makes learning more efficient than CD-PercLoss. For the rest of the paper, we focus on MP training method.

In Figure 2, we visualize the generated samples conditioned on one input modality (e.g., left or right halves of digits). There are many samples generated by the models with MinVI objective that look clearly better than those generated by the model with ML objective.

### 4.2 MIR-Flickr Database

In this section, we evaluate our methods on MIR-Flickr database [11], which is composed of 1 million examples of image and their user tags collected from the social photo-sharing website Flickr.[5] Among those, 25000 examples are annotated with 24 potential topics and 14 regular topics, which leads to 38 classes in total with distributed class membership. The topics include object categories such as dog, flower, and people, or scenic concepts such as sky, sea, and night.

We used the same visual and text features as in [27].[6] Specifically, the image feature is 3857 dimensional vector composed of Pyramid Histogram of Words (PHOW) features [3], GIST [22], and MPEG-7 descriptors [18]. We preprocessed the image features to have zero mean and unit variance for each dimension across all examples. The text feature is a word count vector of 2000 most frequent tags. The number of tags varies from 0 to 72, with 5.15 tags per example in average.

Following the experimental protocol [12, 27], we randomly split the labeled data into 15000 for training and 10000 for testing, and used 5000 from training set for validation. We iterate the procedure for 5 times and report the mean average precision (mAP) over 38 classes.

**Model Architecture.** As used in [27], the network is composed of [3857, 1024, 1024] variables for visual pathway, [2000, 1024, 1024] variables for text pathway, and 2048 variables for top-layer MRBM. As described in Section 3, we pretrain the modality-specific deep networks in a greedy

layerwise manner, and finetune the whole network by initializing MDRNN with the pretrained network. Specifically, we used gaussian RBM for the bottom layer of visual pathway and binary RBM for text pathway.[7] The intermediate layers are trained with binary RBMs, and the top-layer MRBM is trained using MP training algorithm. For the layer-wise pretraining of RBMs, we used PCD [30] to approximate gradient. Since our algorithm requires both data modalities during the training, we excluded examples with too sparse or no tags from unlabeled dataset and used about 750K examples with at least 2 tags. After unsupervised training, we extract joint feature representations of the labeled training data and use them to train multiclass logistic regression classifiers.

**Recognition Tasks.** For recognition tasks, we train multiclass logistic regression classifiers using joint representations as input features. Depending on the availability of data modalities at testing time, we evaluate the performance using multimodal queries (i.e., both visual and text data are available) and unimodal queries (i.e., visual data is available while the text data is missing). The summary results are in Table 2. We report the test set mAPs of our proposed model and compared to other methods. The proposed MDRNN outperformed the previous state-of-the-art in multimodal queries by $4.5\%$ in mAP. The performance improvement becomes more significant for unimodal queries, achieving $7.6\%$ improvement in mAP over the best published result. As we used the same input features in [27], the results suggest that our proposed algorithm learns better representations shared across multiple modalities.

| Model | Multimodal query |
|---|---|
| Autoencoder | 0.610 |
| Multimodal DBM [27] | 0.609 |
| Multimodal DBM$^{\dagger}$ [28] | 0.641 |
| MK-SVM [7] | 0.623 |
| TagProp [31] | 0.640 |
| **MDRNN** | $\mathbf{0.686} \pm 0.003$ |

| Model | Unimodal query |
|---|---|
| Autoencoder | 0.495 |
| Multimodal DBM [27] | 0.531 |
| MK-SVM [7] | 0.530 |
| **MDRNN** | $\mathbf{0.607} \pm 0.005$ |

Table 2: Test set mAPs on MIR-Flickr database. We implemented autoencoder following the description in [21]. Multimodal DBM$^{\dagger}$ is supervised finetuned model. See [28] for details.

To take a closer look into our model, we performed additional control experiment. In particular, we explore the benefit of recurrent encoding network structure of MDRNN. We compare the performance of the models with different number of mean-field iterations.[8] We report the validation set mAPs of models with different number of iterations ($0 \sim 10$) in Table 3. For multimodal query, the MDRNN with $10$ iterations improves the recognition performance by only $0.8\%$ compared to the model with $0$ iterations. However, the improvement becomes significant for unimodal query, achieving $5.0\%$ performance gain. In addition, we note that the largest improvement was made when we have at least one iteration (from 0 to 1 iteration, $3.4\%$ gain; from 1 to 10 iteration, $1.6\%$ gain). This suggests that the most crucial factor of improvement comes from the inference with reconstructed missing data modality (e.g., text features), and the quality of inferred missing modality improves as we increase the number of iterations.

| # iterations | 0 | 1 | 2 | 3 | 5 | 10 |
|---|---|---|---|---|---|---|
| Multimodal query | 0.677 | 0.678 | 0.679 | 0.680 | 0.682 | **0.685** |
| Unimodal query | 0.557 | 0.591 | 0.599 | 0.602 | 0.605 | **0.607** |

Table 3: Validation set mAPs on MIR-Flickr database with different number of mean-field iterations.

**Retrieval Tasks.** We perform retrieval tasks using multimodal and unimodal input queries. Following the experimental setting in [27], we select 5000 image-text pairs from the test set to form a database and use 1000 disjoint set of examples from the test set as queries. For each query example, we compute the relevance score to the data points as a cosine similarity of joint representations. The binary relevance label between query and the data points are determined $1$ if any of the 38 class labels are overlapped. Our proposed model achieves **0.633** mAP with multimodal query and **0.638** mAP with unimodal query. This significantly outperforms the performance of multimodal DBM [27], which reported $0.622$ mAP with multimodal query and $0.614$ mAP with unimodal query.

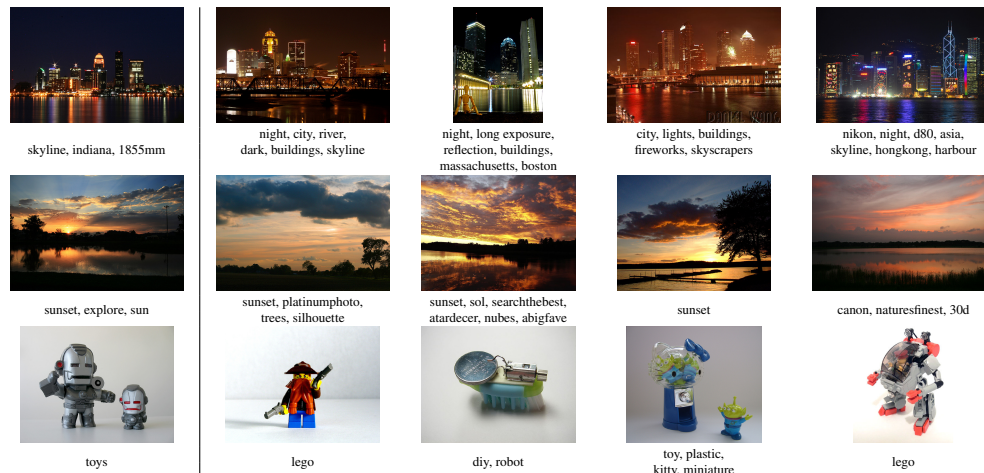

Figure 3: Retrieval results with multimodal queries. The leftmost image-text pairs are multimodal query samples and those in the right side of the bar are retrieved samples with the highest similarities to the query sample from the database. We include more results in supplementary material.

## 4.3   PASCAL VOC 2007

We evaluate the proposed algorithm on PASCAL VOC 2007 database. The original dataset doesn't contain user tags, but Guillaumin et al. [7] has collected the user tags from Flickr website.[9]

Motivated from the success of convolutional neural networks (CNNs) on large-scale visual object recognition [14], we used the $DeCAF_7$ features [5] as an input features for visual pathway, where $DeCAF_7$ is $4096$ dimensional feature extracted from the CNN trained on ImageNet [4]. For text features, we used the vocabulary of size $804$ suggested by [7]. For unsupervised feature learning of MDRNN, we used unlabeled data of MIR-Flickr database while converting the text features using the new vocabulary from PASCAL database. The network architecture used in this experiment is as follows: $[4096, 1536, 1536]$ variables for the visual pathway, $[804, 512, 1536]$ variables for the text pathway, and $2048$ variables for top-layer joint network.

Following the standard practice, we report the mAP over 20 object classes. The performance improvement of our proposed method was significant, achieving $\mathbf{81.5}\%$ mAP with multimodal queries and $\mathbf{76.2}\%$ mAP with unimodal queries, whereas the performance of baseline model was $74.5\%$ mAP with multimodal queries ($DeCAF_7$ + Text) and $74.3\%$ mAP with unimodal queries ($DeCAF_7$).

## 5   Conclusion

Motivated from the property of good generative models of multimodal data, we proposed a novel multimodal deep learning framework based on variation of information. The minimum variation of information objective enables to learn a good shared representations of multiple heterogeneous data modalities with a better prediction of missing input modality. We demonstrated the effectiveness of our proposed method on multimodal RBM and its deep extensions and showed state-of-the-art recognition performance on MIR-Flickr database and competitive performance on PASCAL VOC 2007 database with multimodal (visual + text) and unimodal (visual only) queries.

### Acknowledgments

This work was supported in part by ONR N00014-13-1-0762, Toyota, and Google Faculty Research Award.

## Footnotes

[1]In practice, we use finite samples of the training data and use a regularizer (e.g., $l_2$ regularizer) to avoid overfitting to the finite sample distribution.

[2] For energy-based models like RBM and DBM, it is straightforward to see that every state has non-zero probability and can be reached from any other state. However, the mixing of the chain might be slow in practice.

[3]In CD-PercLoss learning, we run separate Gibbs chains for different conditioning variables and select the negative particles with the lowest free energy among sampled particles. We refer [19] for further details.

[4]There could be different ways of constructing MDRNN; for instance, one can construct the RNN with DBM-style mean-field updates. In our empirical evaluation, however, running full pass of bottom-up and top-down updates performed the best, and DBM-style updates didn't give competitive results.

[5]http://www.flickr.com

[6]http://www.cs.toronto.edu/~nitish/multimodal/index.html

[7]We assume text features as binary, which is different from [27] where they modeled using replicated-softmax RBM [25]. The rationale is that the tags are not likely to be assigned more than once for single image.

[8]In [21], they proposed the "video-only" deep autoencoder whose objective is to predict audio data and reconstruct video data when only video data is given as an input during the training. Our baseline model (MDRNN with 0 iterations) is similar, but different since we don't have a reconstruction training objective.

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
