[Supplementary Material]

# Supplementary Material: Improved Multimodal Deep Learning with Variation of Information

**Kihyuk Sohn, Wenling Shang and Honglak Lee**
University of Michigan Ann Arbor, MI, USA
{kihyuks,shangw,honglak}@umich.edu

## S1 Derivation of Equation (4)

The NLL objective function can be written as

$$
\begin{aligned}
2\mathcal{L}^{\mathrm{NLL}}(\theta) &= -2\mathbb{E}_{P_{\mathcal{D}}}\big[\log P_\theta(X,Y)\big] \\
&= -\mathbb{E}_{P_{\mathcal{D}}}\big[\log P_\theta(X|Y) + \log P_\theta(Y)\big] - \mathbb{E}_{P_{\mathcal{D}}}\big[\log P_\theta(Y|X) + \log P_\theta(X)\big] \\
&= -\mathbb{E}_{P_{\mathcal{D}}}\big[\log P_\theta(X|Y) + \log P_\theta(Y|X)\big] - \mathbb{E}_{P_{\mathcal{D}}}\big[\log P_\theta(X) + \log P_\theta(Y)\big] \\
&= \mathcal{L}^{\mathrm{VI}}(\theta) - \mathbb{E}_{P_{\mathcal{D}}}\big[\log P_\theta(X)\big] - \mathbb{E}_{P_{\mathcal{D}}}\big[\log P_\theta(Y)\big] &\text{(S1)}
\end{aligned}
$$

$$
= \mathcal{L}^{\mathrm{VI}}(\theta) + \underbrace{\mathbb{E}_{P_{\mathcal{D}}}\left[\log\frac{P_{\mathcal{D}}(X)}{P_\theta(X)}\right]}_{KL(P_{\mathcal{D}}(X)\|P_\theta(X))} + \underbrace{\mathbb{E}_{P_{\mathcal{D}}}\left[\log\frac{P_{\mathcal{D}}(Y)}{P_\theta(Y)}\right]}_{KL(P_{\mathcal{D}}(Y)\|P_\theta(Y))} \tag{S2}
$$

$$
\underbrace{-\mathbb{E}_{P_{\mathcal{D}}}\big[\log P_{\mathcal{D}}(X)\big] - \mathbb{E}_{P_{\mathcal{D}}}\big[\log P_{\mathcal{D}}(Y)\big]}_{C_1}
$$

$$
= \mathcal{L}^{\mathrm{VI}}(\theta) + KL\left(P_{\mathcal{D}}(X)\|P_\theta(X)\right) + KL\left(P_{\mathcal{D}}(Y)\|P_\theta(Y)\right) + C_1 \tag{S3}
$$

where Equation (S1) holds by the definition of $\mathcal{L}^{\mathrm{VI}}(\theta)$. Note that $C_1$ is independent of $\theta$. Similarly, we can rewrite the MinVI objective as

$$
\mathcal{L}^{\mathrm{VI}}(\theta) = -\mathbb{E}_{P_{\mathcal{D}}}\big[\log P_\theta(X|Y) + \log P_\theta(Y|X)\big] \tag{S4}
$$

$$
= \mathbb{E}_{P_{\mathcal{D}}}\left[\log\frac{P_{\mathcal{D}}(X|Y)}{P_\theta(X|Y)}\right] + \mathbb{E}_{P_{\mathcal{D}}}\left[\log\frac{P_{\mathcal{D}}(Y|X)}{P_\theta(Y|X)}\right] \tag{S5}
$$

$$
\underbrace{-\mathbb{E}_{P_{\mathcal{D}}}\big[\log P_{\mathcal{D}}(X|Y)\big] - \mathbb{E}_{P_{\mathcal{D}}}\big[\log P_{\mathcal{D}}(Y|X)\big]}_{C_2}
$$

where in Equation (S5), we have

$$
\mathbb{E}_{P_{\mathcal{D}}}\left[\log\frac{P_{\mathcal{D}}(X|Y)}{P_\theta(X|Y)}\right] = \sum_y P_{\mathcal{D}}(y)\mathbb{E}_{P_{\mathcal{D}}(X|y)}\left[\log\frac{P_{\mathcal{D}}(X|y)}{P_\theta(X|y)}\right] \tag{S6}
$$

$$
= \mathbb{E}_{P_{\mathcal{D}}(Y)}\big[KL\left(P_{\mathcal{D}}(X|Y)\|P_\theta(X|Y)\right)\big] \tag{S7}
$$

Finally, we have

$$
\begin{aligned}
\mathcal{L}^{\mathrm{VI}}(\theta) = \mathbb{E}_{P_{\mathcal{D}}(X)}\big[KL\left(P_{\mathcal{D}}(Y|X)\|P_\theta(Y|X)\right)\big] + \\
\mathbb{E}_{P_{\mathcal{D}}(Y)}\big[KL\left(P_{\mathcal{D}}(X|Y)\|P_\theta(X|Y)\right)\big] + C_2.
\end{aligned} \tag{S8}
$$

$C_2$ is independent of $\theta$ and by setting $C = C_1 + C_2$, we derive the Equation (4).

## S2  Proof of Theorem 2.1

**Proposition S2.1** ([1, 2]). *Let $\mathcal{X}$ be a finite state space. Let irreducible transition matrices $T_n$ and $T$ converge to $\pi_n(X)$ and $\pi(X)$, respectively, where $\pi(X) = P_\mathcal{D}(X)$ is a data-generating distribution of $X$. If $T_n$ converges to $T$ in the induced matrix norm, which is denoted by $\|\cdot\|$, then $\pi_n(X)$ converges to $P_\mathcal{D}(X)$ in $l^2$ norm.*

*Proof.* Let $|\mathcal{X}|$ be the number of states. For simplicity, we denote $\pi = \pi(X)$ and $\pi_n = \pi_n(X)$. Since $\pi$ is a stationary distribution of irreducible transition matrix $T$, $\pi$ is uniquely defined and it satisfies the following:

$$T\pi = \pi, \; \mathbf{1}^\top \pi = 1. \tag{S9}$$

Combining above two equations, we have

$$\underbrace{\begin{bmatrix} T_{1,1}-1 & T_{1,2} & \cdots & T_{1,|\mathcal{X}|} \\ T_{2,1} & T_{2,2}-1 & \cdots & T_{2,|\mathcal{X}|} \\ \vdots & \cdots & \cdots & \vdots \\ T_{|\mathcal{X}|-1,1} & \cdots & \cdots & T_{|\mathcal{X}|-1,|\mathcal{X}|-1}-1 \\ 1 & 1 & \cdots & 1 \end{bmatrix}}_{=\widetilde{T}} \pi = \begin{bmatrix} 0 \\ 0 \\ \vdots \\ 1 \end{bmatrix} \tag{S10}$$

Since $\pi$ exists and unique, $\widetilde{T}$ is invertible and we have

$$\pi = \widetilde{T}^{-1} \begin{bmatrix} 0 & 0 & \cdots & 1 \end{bmatrix}^\top \tag{S11}$$

and similarly,

$$\pi_n = \widetilde{T}_n^{-1} \begin{bmatrix} 0 & 0 & \cdots & 1 \end{bmatrix}^\top \tag{S12}$$

Since $T_n$ (entrywise) converges to $T$, $T_n^{-1}$ also converges to $T^{-1}$. Therefore, we conclude $\pi_n$ converges to $\pi = P_\mathcal{D}(X)$. $\qquad\square$

Now, we provide a proof of Theorem 2.1.

*Proof of Theorem 2.1.* To prove the convergence of marginal distributions, it is sufficient to show the convergence of transition operators. Since $|\mathcal{X}|$ and $|\mathcal{Y}|$ are finite, for any $\epsilon > 0$, there exists $N$ such that $\forall n \geq N$, with probability at least $1 - \epsilon$, $\forall x \in \mathcal{X}, \forall y \in \mathcal{Y}$,

$$|P_{\theta_n}(y|x) - P_\mathcal{D}(y|x)| < \epsilon, \; |P_{\theta_n}(x|y) - P_\mathcal{D}(x|y)| < \epsilon$$

The transition operators are defined as follows:

$$T_n^\mathcal{Y}(y[t]|y[t-1]) = \sum_{x \in \mathcal{X}} P_{\theta_n}(y[t]|x) P_{\theta_n}(x|y[t-1]),$$

$$T^\mathcal{Y}(y[t]|y[t-1]) = \sum_{x \in \mathcal{X}} P_\mathcal{D}(y[t]|x) P_\mathcal{D}(x|y[t-1])$$

where $P_{\theta_n}(x|y)$ and $P_{\theta_n}(y|x)$ are derived from the joint distribution $P_{\theta_n}(x,y)$ and similarly for data-generating distribution, $P_\mathcal{D}(x|y)$ and $P_\mathcal{D}(y|x)$ are derived from $P_\mathcal{D}(x,y)$. Then, for $n \geq N$, we have, for any $y_t, y_{t-1} \in \mathcal{Y}$, with probability at least $1 - \epsilon$,

$$\left| T_n^\mathcal{Y}(y_t|y_{t-1}) - T^\mathcal{Y}(y_t|y_{t-1}) \right|$$

$$\leq \left| \sum_{x \in \mathcal{X}} P_{\theta_n}(y_t|x) P_{\theta_n}(x|y_{t-1}) - P_\mathcal{D}(y_t|x) P_\mathcal{D}(x|y_{t-1}) \right|$$

$$\leq |\mathcal{X}| \max_{x \in \mathcal{X}} \left| P_{\theta_n}(y_t|x) P_{\theta_n}(x|y_{t-1}) - P_\mathcal{D}(y_t|x) P_\mathcal{D}(x|y_{t-1}) \right| \tag{S13}$$

$$\leq |\mathcal{X}|(2\epsilon)$$

As we assume finite sets $\mathcal{X}$ and $\mathcal{Y}$, this proves the convergence (in probability) of transition operator $T_n^\mathcal{Y}$ to $T^\mathcal{Y}$. The same argument holds for the convergence of transition operator $T_n^\mathcal{X}$ to $T^\mathcal{X}$. With

Proposition S2.1, we proved the convergence of asymptotic marginal distribution $\pi_n(X)$ and $\pi_n(Y)$ to data-generating marginal distributions $P_{\mathcal{D}}(X)$ and $P_{\mathcal{D}}(Y)$, respectively.

Now, let's look at the joint probability distributions $P_{\theta_n}(x, y) = P_{\theta_n}(x|y)P_{\theta_n}(y)$ and similarly, $P_{\mathcal{D}}(x, y) = P_{\mathcal{D}}(x|y)P_{\mathcal{D}}(y)$. As we proved above, the following inequalities hold $\forall n \geq N'$:

$$\left| P_{\theta_n}(y) - P_{\mathcal{D}}(y) \right| < \epsilon, \ \left| P_{\theta_n}(x|y) - P_{\mathcal{D}}(x|y) \right| < \epsilon \tag{S14}$$

Therefore, using the similar argument in Equation (S13), we have

$$\left| P_{\theta_n}(x, y) - P_{\mathcal{D}}(x, y) \right| < 2\epsilon \tag{S15}$$

and this completes the proof. $\qquad\square$

# S3 Retrieval Task

We provide more results of retrieval task with multimodal queries on MIR-Flickr database.

sky, night, clouds, space

sky, night, stars

sky, night, mountains, stars, fab

2007

nikon, nature, sky, night, landscape, impressedbeauty, d300, dark, longexposure, colorado, stars

2007, beauty, hair, friend

bw, portrait, blackandwhite, girl, nikon80

blackandwhite, selfportrait, happy, mac, makeup

bw, portrait, photo

2007, may

studio, craft, room

home, toys, interior, bed, books, decor

chair

design, studio

studio, craft

puppy

cute, puppy

puppy

dog

explore

skyline, indiana, 1855mm

night, city, river, dark, buildings, skyline

night, reflection, longexposure, buildings, massachusetts, boston

city, lights, buildings, fireworks, skyscrapers

nikon, night, d80, asia, skyline, hongkong, harbour

portrait, me

portrait, man, colours

selfportrait, me, 365days, 365, self

selfportrait, 365days, 365

portrait, girl, woman, birthday

portrait, explore, blackwhite, portraits

portrait

london, uk

bw, halloween

blackandwhite, milan

white, me

bw, selfportrait, me, layers

bw, blackandwhite, selfportrait, me, 365days, photoshop, self, face, head, myself, me

de

bw, self

blue, night, city, explore, nyc, newyork, lights, newyorkcity, manhattan, ny, skyline, cityscape, twilight, skyscrapers

nyc, newyorkcity

city

night, lights, new, york, exposure, noche, long, pier

sunset, chicago, tower, skyline, dusk, skyscraper

sunset, explore, sun

sunset, trees, platinumphoto, silhouette

atardecer, sunset, abigfave, searchthebest, nubes, sol

sunset

canon, naturesfinest, 30d

home, modern, chair | chair | california, home, design, ca, day, interior, rainbow, chair, books, library, apartment, decor | home | desk

knitting | desk | home | design, studio | window, house, door, books

apple, ipod | iphone, macbook | nyc, newyorkcity, pro | canoneos350d | iphone

car, ford, gt, estate | car, british | carshow | car | carshow

toys | lego | diy, robot | toy, kitty, plastic, miniature | lego

blackandwhite, selfportrait, bw, 365days, symmetry | selfportrait, 365days, blackandwhite, dancing | bw, portrait, nikon40, hands | bw, chile, mujer | light, window, blackandwhite

portrait, blackandwhite, girl, newyork, best | nikon, bw, portrait, blackandwhite, 365days, music, self, friends, d50, hair | bw, portraits | bw, portrait, japan, beautiful, fuji, face | bw, blackandwhite, bn, milan

portrait, 365days, 365, self | november | selfportrait, 365days, 365 | self | me

laptop | work, mac, lighting, computer, desk | books, diy | pen | iphone, apple, mac, ipod, macbook

graffiti, streetart, chile, rio | graffiti, portugal, streetart, lisboa, lisbon | graffiti, 2007, graf, tags, graff | graffiti, 2007, graf, tags, graff | graffiti, nyc, streetart

Figure S1: Retrieval results with multimodal queries on MIR-Flickr database. The leftmost image-text pairs are multimodal queries and those in the right side of the bar are retrieved samples with the highest similarities to the query.