[Reviews · NeurIPS 2014]

Submitted by Assigned_Reviewer_25

Summary:
This paper proposes a different way of training multimodal deep models. For
inputs X and Y, the model tries to maximize the sum of two conditional log probs -
P(X|Y) and P(Y|X). This is in contrast to other models like DBMs that try to
maximize the joint log prob P(X, Y).

Strengths:
- This paper tries to address a very important question about which is the better way of dealing with multiple modalities : training conditional models or joint models.
- The model gets impressive results on multiple datasets.

Weaknesses:
- One of the points that this paper tries to make is that learning two
conditional models is easier than learning a joint one. While this is definitely
plausible, some more comparisons need to be made to empirically verify this
claim.

(1) The model should be compared to a simpler autoencoder-style model that has $x$
and $y$ as inputs and is trained to reconstruct both of them (similar to the
approach of Ngiam et al. (ICML 2011)). That seems like a natural comparison to
make since such an autoencoder can be seen as a one-step unrolling of the kind of
computation that this model unrolls several times. For unimodal tasks, the unknown modality
can simply be set to zero.

(2) Another comparison that would help is to again have both $x$ and $y$ as
inputs and train a model that is similar to the proposed MDRNN except that it predicts both $x$ and $y$ after going through several iterations of unrolling.

Having these comparisons in the paper would probably make a strong point about whether we should be training jointly on multiple modalities or conditioning on one.

- In Table 3, it would be great to include results for number of iterations between 0 and 10. Maybe, 1 and 5 ? The table, as it is now, is enough to convince us that having some iterations is better than having none, but it would be even better if it also gave a sense how much we gain by iterating 10 times as opposed 1 or 5.
Summary: The paper explores an important question of whether conditional or joint models are better suited to modeling multiple modalities. The results are impressive, however, having some additional experimental comparisons would make this paper even more compelling.

Submitted by Assigned_Reviewer_37

The paper describes a new training objective for multi-modal networks. The new objective is quite simple and can be applied to many different graphical models and has been shown in the paper to be related to maximum likelihood. The idea is to minimize the variation of information between the two modalities. This amounts to minimizing the cross-entropy of the conditional distributions going from one modality to the other. The idea is applied to training RBMs and RNNs (through multi-prediction training). The paper is well-written and clear.

Section 2 is very interesting in that it establishes a clear link with maximum likelihood. It shows exactly what is lost by using only the VI as the learning objective. This allows us to understand better how and when the method will be successful.

The paper would benefit largely from a section reviewing the past work on multi-modal learning. It could easily replace the section 3.1 which covers very basic features of the RBM.

L199. typo "to be good"

L341. "The intermediate layers are trained with binary RBMs, the top layers with the MP algorithms". It is not clear from this phrase if CD or PCD, or some other method was used to train the binary RBMs.
Summary: A simple but effective approach to multi-modal learning with neural networks. The approach is validated successfully on a range of datasets.

Submitted by Assigned_Reviewer_38

This paper proposes a modality reconstruction criterion (given the other
modalities) for training the top level of a multi-modal architecture (with
the lower levels pre-trained by unsupervised learning on a per-modality
basis, and a separate path for each modality, like in previous multi-modal
modeling with the DBM).

The mathematical results are pretty straightforward and unsurprising, and
one would think that this methodology is less "powerful" than the maximum
likelihood training, of say DBMs (which have a similar architecture). Yet
the comparative experimental results are impressive and beat the DBM
consistently across many benchmarks. The fact that the whole thing is
fine-tuned like in the MP-DBM may be part of the explanation (and it would
be good to test it, i.e., what happens without the fine-tuning part?).

One thing I don't like, besides the fact that the training criterion is not
trying to model the marginals, is that the conditionals are approximated by
a factorial distribution, which seems like a very strong assumption. I
suspect that the reason it seems to work so well is that the tasks involve
choosing one value of a modality given the value observed for the other. So
in a sense we are not even interested in the posterior distribution but
more in the MAP, and a unimodal factorial posterior is then a reasonable
way to focus on an estimated MAP. I bet that sampling from one modality
given the other would not give so nice results (unlike the case of
the DBM). But maybe that's ok, and if we really want to solve the
kinds of problems in the benchmark, one is really better off training
in this MAP-like manner than in the full probabilistic mode that
DBMs enjoy.

So I think that although the ideas in this paper are simple on the
surface, the very surprisingly good results and the questions these
raise are very interesting and worth publishing. I would modify the
paper accordingly, trying to focus the attention, the discussion,
and maybe extra experiments, in trying to understand *why* the
proposed approach works better than the multi-modal DBM.

Toy experiments on MNIST: Note that CD is not recognized to be a
particularly good way of training RBMs in a generative modeling perspective, so
you should really try to train them by PCD/SML instead. And you can't
really call CD-training "maximum likelihood" because it is quite far from
it. PCD/SML is closer to maximum likelihood.

Spell out 'GT' in Figure 2.
Summary: Summary of review: reasonable but rather straightforward approach to
training multi-modal deep nets based on a conditional modality
reconstruction criterion, which suprisingly beat multimodal DBMs across
many benchmarks. The good results may be tied to the specifics of the
tasks (reconstructing one modality given the other). Some of the
comparisons could be improved (CD is not recognized as the best way to
train generative RBMs) but the fact that on the "real-world" benchmarks
the comparative results are very good is a big plus.
Author Feedback
Author rebuttal: Thank you for the valuable comments and suggestions. We will reflect all your comments and revise the paper to provide more clarifications and insights of our proposed method.

R25, R38:
Based on your suggestions, we have performed additional control experiments. For all these control experiments (on the Flickr database), we evaluated with the same network architecture as described in Section 4.2 in the paper.

R25:
Regarding comparison to autoencoder-style models, we evaluated the performance of bimodal deep autoencoder [16] on Flickr database, and obtained 0.610 and 0.495 mAPs on the test set for multimodal and unimodal queries, respectively. (Note that [20] reported similar results of 0.600 mAP for multimodal queries.) These results are significantly worse than 0.686 and 0.607 mAPs for multimodal and unimodal queries based on our method (MDRNN). This shows that our method significantly improves over the simple baseline based on autoencoder.

We compared the recognition performance of MDRNN with different number of RNN iterations during training (the same number of iterations used for training was used for testing). The summary results (validation set mAPs) are given below (0, 1, 5, 10 iterations, respectively)*:

Multimodal query (mAP): 0.677, 0.678, 0.682, 0.685
Unimodal query (mAP): 0.557, 0.591, 0.605, 0.607

(* “0 RNN iteration” means feedforward computation without any loops.)

The most significant improvement was made for unimodal query when we increase the iteration number from 0 to 1 (margin of improvement: 0.034), but we still observe further gain by increasing it to 5 and 10 (margin of improvement: 0.048 and 0.05).

Based on your suggestion on multi-prediction training that approximates joint likelihood, we evaluated the performance of the multimodal deep network trained jointly on $x$ and $y$ like in the original MP-DBM (i.e., randomly select subsets of variables from both data modalities $x$ and $y$ and predict them given the rest). In our preliminary results, the original MP-DBM style training jointly on $x$ and $y$ gave worse results than our proposed training scheme (i.e., predicting $x$ given $y$ and vice versa) for both multimodal and unimodal queries. We will include complete results in the revision.

R38:
Fine-tuning brings a significant improvement: before MDRNN fine-tuning, we obtained 0.632 and 0.521 test set mAPs for multimodal and unimodal queries, respectively, and these numbers go up to 0.686 and 0.607 mAPs after MDRNN fine-tuning. Interestingly, the performance of our model without fine-tuning is comparable or better than fine-tuned results from multimodal DBM [20] (0.609 and 0.531 mAPs for multimodal and unimodal queries). This result suggests the advantage of our training method based on conditional objectives.

We agree that it may be a strong assumption to approximate the conditionals by a factorial distribution. However, as you suggested, we found that this was not very problematic in practice since we condition on one input modality. We plan to investigate ways to approximate the conditionals via non-factorial distributions for future work.

R38:
We will modify the paper by adding discussions and additional experimental results to provide insights on why our proposed method works better than the multi-modal DBM. For this purpose, we also plan to work on theoretical analysis of our method.

R37, R38:
We are sorry for the confusion and typos regarding descriptions about training methods for RBMs. We trained RBMs using PCD (Tieleman, 2008)** for all cases when applicable (e.g., model with ML objective in Section 4.1, layer-wise pretraining of multimodal deep networks in Section 4.2 and 4.3). In other words, in Table 1, “ML (CD)” denotes results obtained from PCD training. In Section 4.2 and 4.3, we trained the Gaussian and binary RBMs using PCD for layer-wise pretraining. We will correct and clarify these in the revision.

** Tieleman, Training Restricted Boltzmann Machines using Approximations to the Likelihood Gradient, ICML 2008

R37:
We will revise the paper by adding a section reviewing the past work on multi-modal learning.